# The Use of Artificial Intelligence for Skin Cancer Detection in Asia—A Systematic Review

**DOI:** 10.3390/diagnostics15070939

**Published:** 2025-04-07

**Authors:** Xue Ling Ang, Choon Chiat Oh

**Affiliations:** 1Department of Internal Medicine, Singapore Health Services, Singapore 169608, Singapore; xueling.ang@mohh.com.sg; 2Department of Dermatology, Singapore General Hospital, Singapore 169608, Singapore; 3Duke-NUS Medical School, Singapore 169857, Singapore

**Keywords:** artificial intelligence, machine learning, skin cancers, malignancy, Asian, skin of color

## Abstract

**Background**: Artificial intelligence (AI) developed for skin cancer recognition has been shown to have comparable or superior performance to dermatologists. However, it is uncertain if current AI models trained predominantly with lighter Fitzpatrick skin types can be effectively adapted for Asian populations. **Objectives**: A systematic review was performed to summarize the existing use of artificial intelligence for skin cancer detection in Asian populations. **Methods**: Systematic search was conducted on PubMed and EMBASE for articles published regarding the use of artificial intelligence for skin cancer detection amongst Asian populations. Information regarding study characteristics, AI model characteristics, and outcomes was collected. **Conclusions**: Current studies show optimistic results in utilizing AI for skin cancer detection in Asia. However, the comparison of image recognition abilities might not be a true representation of the diagnostic abilities of AI versus dermatologists in the real-world setting. To ensure appropriate implementation, maximize the potential of AI, and improve the transferability of AI models across various Asian genotypes and skin cancers, it is crucial to focus on prospective, real-world-based practice, as well as the expansion and diversification of existing Asian databases used for training and validation.

## 1. Introduction

Skin cancer incidence and clinical presentations differ between Asian and Caucasian populations due to variations in Fitzpatrick skin phototype [1], lifestyle, and genetic background [2]. Non-melanoma skin cancers (NMSC) such as basal cell carcinomas (BCC) and squamous cell carcinomas (SCC) are historically less prevalent amongst Asian populations as darker skin types are associated with increased epidermal melanin and melanocyte activity, which confers greater protection against ultraviolet B (UVB) radiation [3]. In contrast to Caucasian populations, BCCs tend to affect older Asians and may occur on less typical sites such as the trunk or as a more pigmented lesion [3]. Likewise in Asia, the incidence rate of cutaneous melanoma is significantly lower than in Caucasian populations, and the most common histological subtype, acral lentiginous melanoma (ALM), which accounts for roughly 50% of all melanoma cases, only constitutes about 2–3% of cases in Caucasian populations [4]. Despite lower prevalence of melanoma, the mortality rate is often higher in Asian populations, whereby patients are typically diagnosed at a later stage of disease [5,6].

Even within Asia or a multi-ethnic country, like Singapore [1,4], skin cancer prevalence can differ significantly as well due to heterogeneity in the intrinsic skin phototype, genetics, lifestyle, environmental factors, and disease awareness [7]. For instance, basal cell carcinoma is the most prevalent among Chinese and Japanese individuals, while squamous cell carcinoma is more common among people of Indian descent [3]. The rates of skin cancers have also been found to differ amongst the ethnic Chinese, Indians, and Malays in Singapore [1]. Socio-economic background may also contribute to differences in incidence rate, as skin cancer detection rates might be higher in urban or resource-rich countries where there is greater access to the healthcare system [6].

The global rise in skin cancer incidence [8] and increasing recognition of disparity in healthcare access [9,10] have led to a search for solutions to aid in earlier skin cancer detection and improvement of healthcare equity.

Artificial intelligence (AI) refers to the development of intelligent computer systems that can simulate human behavior and cognition to perform tasks independently [11]. In recent years, the advent of large-scale datasets and computational power has propelled the transition of AI from conventional rule-based systems to sophisticated machine learning [12], which consists of deep learning models, such as artificial neural networks (ANNs) and convolutional neural networks (CNNs) [13,14]. ANNs are non-linear statistical prediction models, which utilize multi-layer neural networks to learn the complex associations between data, while CNNs are a subtype of ANN that specializes in image recognition and classification [15]. In the last two years, there has also been an increasing trend to tap on transformers and large language models (LLMs), especially multi-modal LLMs, composed of multiple transformer layers, which allows the algorithms to selectively focus on different parts of the input data to analyze and interpret both visual and textual information [16].

As compared to the use of non-invasive devices such as confocal and photo microscopy for skin cancer detection, the development of these automated visual recognition AI models [17,18] is especially suitable for visual-based specialties, such as dermatology, and can allow even inexperienced practitioners to mitigate the limitations associated with real-life consultations and healthcare inequality [19,20] to provide more accurate diagnoses, which not only benefit patients but also relieve healthcare systems.

While previous systematic reviews [21,22] have shown promising results with the use of AI for skin cancer detection, the current literature mainly constitutes research and databases derived from Caucasian populations. This limits understanding of regional variations and trends in AI development for skin cancer detection and raises concerns as to whether successful AI models implemented thus far can be adapted for Asian contexts given innate differences between Caucasian and Asian populations.

Therefore, our systematic review will only focus on relevant studies utilizing Asian population databases for the development or validation of AI models to (1) summarize the current use of artificial intelligence for skin cancer detection in various Asian countries, (2) identify trends and challenges faced, and (3) propose areas for improvement.

## 2. Materials and Methods

The PubMed and EMBASE databases were searched for relevant articles without restrictions up till 27 August 2024. We used the following keywords (“skin neoplasm” OR “skin carcinoma” OR “skin cancer” OR “basal cell carcinoma” OR “squamous cell carcinoma” OR “melanoma” OR “keratinocyte carcinoma”) AND (“artificial intelligence” OR “machine learning” OR “neural networks” OR “computer aided” OR “deep learning”) for our search.

The review protocol was not registered with PROSPERO but was performed in accordance with the PRISMA guidelines and registration information, whereby only primary studies, which evaluated the use of AI for skin cancer detection were included, as shown in Figure 1 below. After duplicates were removed, the authors screened titles and abstracts to identify studies that met the inclusion criteria. Studies that did not specify skin cancer, the use of artificial intelligence, or the use of Asian population databases for training or validation of AI models were excluded to ensure that our review focuses on outcomes achieved with Asian databases. Where studies did not clearly specify details of the database included, we sought information from the database website wherever possible. Of the titles and abstracts, 34 studies were included for full-text screening. Any disagreements were discussed by the authors, and a consensus was reached. Conference papers, letters, or reviews were excluded. Eventually, 22 studies were included in the final review.

Data extraction was undertaken independently by the reviewers into a predesigned data extraction spreadsheet on Microsoft Excel. The following data were extracted from the included studies: (a) characteristics of studies—country of origin of study, source of database, publication year, types of skin cancers included in dataset, modality of assessment (dermoscopic versus clinical images), sample size of each study, gold standard diagnosis; (b) AI model—type of AI model utilized, details of AI model, data input and output processing methods; and (c) outcomes—classification accuracy, sensitivity, and specificity of the AI model, comparison across models, comparison across modalities (clinical versus dermoscopic images), or performance of AI against physicians.

The two reviewers independently assessed the quality of the studies included and the risk of bias using QUADAS-2 [23]. Based on the questions, we classified each QUADAS-2 domain as low (0), high (1), or unknown (2) risk of bias.

## 3. Results

The initial database search from PubMed and EMBASE identified 3113 and 5100 studies, respectively. A total of 1474 duplicates were removed. After screening by titles, abstracts, and full text, only twenty-two studies [24,25,26,27,28,29,30,31,32,33,34,35,36,37,38,39,40,41,42,43,44,45] were included in the final review.

### 3.1. Characteristics of Studies and Datasets

As summarized in Table 1, studies included were mostly from East Asian countries—China (8/22), Japan (4/22), South Korea (8/22), Taiwan (1/22) and Iran (1/22), with a peak in publication rate in the year 2020. Only institutional or private databases were used in the studies included.

Pre-processing of data was employed in 12/22 studies to improve the accuracy of feature extraction and interpretation by AI. Methods include the removal of low-quality or mislabeled images and artifacts (e.g., clothing, hair, markings, and appearance augmented by previous treatment) as well as cropping, resizing, or enhancement of images to ensure that skin lesions are properly captured and fitted to the input requirements of AI algorithms. Some studies utilized unprocessed images to mimic real-life applications whereby photos might not always be taken by professionals or be of high quality, especially in the community setting.

Data augmentation was also performed in 4/22 studies via rotating and flipping the original image at varying angles to increase the number of images available for training and validation. This is commonly adopted by studies with small datasets to increase the robustness of databases for training AI algorithms.

Most studies utilized histopathological diagnosis as the standard for ground truth (11/22). Six studies included skin lesions that were diagnosed clinically—either because the skin lesion was clinically benign with no justification for biopsy in a real-life setting or in cases whereby histopathological diagnosis was not certain and consensus from a few dermatologists was obtained for clinicopathological correlation (6/22). Five studies did not clearly state the basis of determining the gold standard diagnosis.

In terms of skin cancers evaluated, malignant melanoma was the most commonly included skin cancer (16/22), with five papers focusing on the use of AI algorithms for the detection of an acral melanoma subtype (4/22). This is followed by basal cell carcinoma (15/22) and squamous cell carcinoma (7/22). Only four studies included rarer skin cancers such as intraepithelial carcinoma, Kaposi’s sarcoma, or adenocarcinoma of lips (4/22).

### 3.2. Types of AI Models Used

As shown in Table 1, the studies were categorized into either shallow or deep AI techniques based on the complexity of the AI architecture under the model. Shallow techniques are typically easier to train with simpler structures and only a few layers of neural networks. In contrast, deep technique refers to those with complex architectures containing at least three layers of multiple intervening layers, allowing them to better understand the relationships between inputs and outputs [3].

Table 2 summarizes the AI models used and characteristics of datasets included. CNN-based models were the most popular AI models studied, with Inception ResNetV2 being the most highly utilized algorithm. Some studies custom-built their own algorithms, while others utilized transfer learning by re-training certain convolutional layers or incorporating other algorithms into established AI architectures trained with large Caucasian-derived datasets to improve adaptability to Asian population datasets. Common tasks performed by AI algorithms include feature extraction, segmentation, and binary or multiclass classification of skin lesions. While deep AI techniques, such as CNNs, are able to incorporate and perform all of the common tasks, some studies which employed shallow AI techniques required a combination of AI models to perform different aspects of tasks to allow better classification of skin lesions.

In terms of image input, 11/22 used clinical images of skin lesions that were either taken from mobile phones or high-resolution cameras, while 9/22 utilized dermoscopic photographs instead. Only 2/22 studies included dual modality (both clinical and dermoscopic photographs) to assess skin cancer detection ability. Direct comparison of the input modalities did not reveal any significant difference in the accuracy rate of the AI model for melanoma [39] or basal cell carcinoma [42] detection. However, when both modalities were concurrently used as input, the accuracy rate reduced to 76.2% [42].

### 3.3. Study Outcomes

Table 3 summarizes the study outcomes of studies included. Binary or multiclass classification abilities of AI models were commonly assessed, and 10/22 studies reported AI accuracy more than 90% [25,27,29,30,32,34,37,41,44,45]. Out of these ten studies, three were custom-built algorithms, while seven [25,27,29,30,34,37,41] of them utilized the transfer learning approach, which involves the adaptation of knowledge acquired through learning from large datasets in related domains by fine-tuning them on smaller Asian datasets [46].

Amongst convolutional neural network (CNN) models, two studies compared performance between various CNNs. Abbas et al. found that transfer learning with ResNet-18 attained higher accuracy versus AlexNet and the custom-built Deep ConvNet model [25]. Xie et al. showed that Xception outperformed Inception, ResNet50, Inception ResNetV2, and DenseNet121 when applied to Asian populations [30]. Inception involves multi-scale feature extraction with inception modules; ResNet uses residual learning with skip connections that allow very deep networks; DenseNet ensures dense connections between all layers for efficient feature reuse, while Xception uses depth-wise separable convolutions for more efficient computation [47]. In the context of skin cancer detection, the depthwise separable convolutions technology will allow Xception to be more efficient in processing high-resolution complex images with fewer parameters without overwhelming computational resources or overfitting and allow more robustness and generalizability.

Sixteen studies evaluated the binary classification ability of AI as ‘benign’ versus ‘malignant’. Nine studies [24,27,30,31,32,34,35,39,43] showed comparable performance between algorithms and expert dermatologists; two studies [25,36] showed that AI outperformed experts, while four outperformed trainees or non-experts [24,27,40,43] in their ability to differentiate benign from malignant skin cancer lesions. However, Han et al. showed that the accuracy of the first clinical impression generated by AI was inferior to that of physicians [26].

Twelve studies assessed the multiclass classification ability of AI algorithms, which is the ability to distinguish lesions according to specific disease type [26,28,29,32,33,34,35,37,39,40,42,43]. For instance, these AI models can differentiate the skin lesions to identify if they belong to basal cell carcinoma, malignant melanoma, or basal cell carcinoma groups instead of just discerning if they are likely to be benign or malignant. Three studies showed that the multiclass classification ability of AI algorithms based on a single image was comparable to physicians [26,34,42], while two surpassed the performances of dermatologists [35,39].

In terms of the efficacy of AI assistance, Han et al. [28] showed that AI improved the ability of dermatologists in predicting malignancy and making decisions on treatment options as well as multi-disease classification tasks with significant improvement in the mean sensitivity and specificity of dermatologists. However, another study by Han et al. [40] showed that AI assistance mainly benefited non-dermatology trainees with the top accuracy of the AI-assisted group being significantly higher than that of unaided group, but there was no significant change in performance amongst dermatology residents.

## 4. Discussion

We present, to our knowledge, the first systematic review summarizing the existing AI image-based algorithms developed for skin cancer detection, trained or validated exclusively with a predominantly Asian database.

### 4.1. Potentials and Benefits of Using AI for Skin Cancer Detection

Based on current evidence, several studies suggested the comparability of AI’s skin cancer detection ability to experts [24,25,27,30,32,34,35,39,43] and saw an improvement in accuracy rates when AI is used to augment the decision-making of non-dermatologists in real-world settings [40]. As such, AI’s potential promises several folds of benefits not only to patients but also to physicians and the healthcare system.

AI can increase health equity through improving access to skin screenings. This is especially useful in rural areas or resource-scarce areas where it may take longer for a patient to seek medical attention or obtain a biopsy of suspected skin lesions [48]. For primary care physicians with less experience, AI models with the ability to discern benign versus malignant classifiers can help triage skin lesions requiring specialist referral, and multiclass classifiers can help diagnose skin lesions to tailor treatment and improve explainability and trust for clinicians [49]. Beyond primary care settings, especially in times of pandemic where physical consultations are discouraged [50], AI-augmented tele-dermatology can further enhance accessibility by streamlining referrals and reducing waiting time [48]. In the tertiary institution, a good AI model can also offer diagnostic support to physicians when encountering an atypical skin lesion or non-local population [37]—for instance, an AI model developed for non-local skin cancer detection can offer a second opinion to a doctor who might not have much exposure to non-local patients with differing Fitzpatrick skin type.

### 4.2. Comparison of AI Versus Physicians’ Performance in Skin Cancer Detection

In a pragmatic setting, dermatologists are often equipped with additional information to improve diagnostic accuracy through history-taking, access to medical records, and screening for associated signs and symptoms to support diagnosis. While some studies suggest comparable or better performance by AI models in terms of skin cancer image recognition due to AI’s ability to extract intricate features and discern inconspicuous findings to better distinguish similar-looking lesions, it is difficult to comment if this is a true reflection of AI’s overall diagnostic abilities. For instance, the reduction in accuracy rate when both clinical and dermoscopy images were used as input for AI interpretation [41] might suggest AI’s inability to assimilate findings since the additional details are supposed to improve the accuracy of dermatologists. When skin cancers manifest atypically, the lack of pre-training with similar images might limit AI’s diagnostic ability as well.

Physicians are often presented with myriads of skin lesions not limited to the few groups, which current AI algorithms are trained for. Without adequate external validation, it remains unclear if AI can function equally well in the dynamic real-world setting when confounders are present or when a larger range of skin conditions are introduced. In addition, it is also crucial to note that the first clinical impression of physicians was superior to AI [26], and the diagnostic accuracy of trainees dropped after consulting the AI model if the model’s diagnosis was incorrect, highlighting a possible pitfall of using current AI models to augment decision-making [40].

### 4.3. Comparison of AI Models

Unlike traditional machine learning approaches, deep learning (DL) models have more sophisticated feature extraction techniques to identify correlations within data samples to optimize classification accuracy, detection precision, or segmentation performance and extract higher-order representations from data that are not easily discernible through traditional methods [51].

In the last decade, there has been a shift in machine learning, with the focus now on deep learning approaches such as CNNs, transformers, or their variants, which are refinements on traditional ANNs to improve predictive accuracy and reduce the complexity of previous algorithms [18,52]. One such method is CNNs, which are specially engineered for image processing and analysis. CNNs consist of multiple cascading non-linear modeling units called “layers” that filter the input data by filtering redundant information, deciphering correlations, and summarizing critical information into a distilled representation called “features” before mapping the extracted features to target diagnostic labels or outputs [51]. These require less computational time and power but produce higher predictive accuracies than traditional machine learning approaches [52].

More recently, vision transformers (ViTs), which are “attention”-based models capable of selectively focusing on relevant parts of the input data to generate output, have been garnering increasing popularity for image recognition tasks as they offer several key advantages over CNNs. ViTs are capable of capturing long-range relationships and are able to process images by dividing them into smaller patches and encoding them through self-attention mechanisms to capture relationships between tokens [51,53]. In terms of interpretability, ViTs’ attention maps can provide a clear, localized picture of attention, which provides researchers with new insight into how the model makes decisions, unlike traditional CNNs, which typically utilize explainability methods, such as class activation maps (CAM) and Grad-CAM, which provide limited receptive fields, coarse visualizations, and potentially reduces the accuracy of algorithms [53,54]. Studies by Matsoukas and Xin et al., which employed ViTs showing high accuracy rates of >90% for skin image classification, suggest promising performance in skin cancer classification tasks [54,55]. Large language models built on transformer architectures, such as Skin-GPT4, can allow patients to upload photos of their skin lesions and ask direct questions to simulate tele-dermatology services [56].

Despite the benefits of transformers, at the time of our review, there has not been any published studies developed or trained with Asian databases [16,54,55,56]. This could be due to the lack of significant performance improvement over traditional CNN models when dealing with existing small Asian datasets [54].

Other practical aspects such as training time, inference speed, and hardware requirements are crucial to consider when deciding the suitability and ease of adaptation for real-world application. As compared to traditional machine learning models, such as decision trees or SVMs, deep learning models (like CNNs and transformers) have slower inference speed and require longer training time as well as GPUs or TPUs for training and inference [55]. While consistent input images through data pre-processing and segmentation may reduce computational load and achieve faster training and inference time [57], these can be challenging to implement, especially in resource-poor countries.

### 4.4. Current Challenges and Development of AI in Asia

Amongst the twenty-two studies included, most were derived from East Asian countries, with the highest research output from China, South Korea, and Japan. While there have been studies published from other Asian countries as well [58,59], there is currently a lack of representation from Asian countries with generally darker skin tones, such as Southeast Asian countries or India, as some of these studies were also excluded as they did not utilize local Asian databases.

The disparity in terms of research output and database availability amongst Asian countries can possibly be attributed to income disparity, as developing countries often lack the funding, resources, and talent to explore and utilize AI technology [3,20]. Similarly, from the patient perspective, they may be less motivated to seek medical treatment for skin lesions, especially if they are asymptomatic [3], leading to fewer reported cases available for inclusion in local databases for research.

### 4.5. Existing Asian Databases

The current Asian databases identified are mainly owned by private institutions. In terms of skin cancer type, there has been a greater focus on malignant melanoma detection, likely due to the increased awareness of high mortality associated with late diagnosis. This is followed by basal cell carcinoma, which is the most prevalent skin cancer worldwide. Transfer learning with established CNN-based models trained with large-scale Caucasian databases has been shown to have superior accuracy rates as compared to custom-built algorithms trained with small-scale local Asian databases, suggesting the importance of using large-scale validated databases to fine-tune and improve the accuracy of deep learning models. However, the yield of transfer learning might be limited when conventional models developed based on Caucasian databases for malignant melanoma (using the ABCD approach or 7-point scale approach) cannot be readily transferable to skin cancers with unique clinical features or developing at uncommon anatomical sites such as acral melanoma.

The lack of a large-scale Asian database not only hinders the development of existing deep learning AI models, limits innovation but also jeopardizes patient safety. The fact that existing databases are mostly derived from tertiary hospitals also risks selection bias for patients at higher risk of malignancy [9]. Cho et al.’s study, which compared the results of test sets to external validation sets, revealed a reduction in accuracy rate when an AI algorithm was validated against an external dataset [36]. As Asians are not a homogenous population, diversity exists, and hence, it underscores the importance of constructing a publicly available, validated image database to ensure greater representation across Asian skin genotypes and conditions, and to allow for stringent quality control of AI developed.

Clinical imaging was the most popular input method amongst the studies included. This predilection could be due to the relative ease of collecting clinical images and the lack of a publicly available Asian dermoscopic image database, such as the Asian equivalent of the International Skin Image Collaboration (ISIC) database, for use in the training and validation of AI models developed for skin cancer detection [60]. Test performance of deep learning models is known to be influenced by variations in image acquisition and quality. Therefore, while this review found no significant differences in accuracy rates for skin cancer detection across studies using different modalities, such comparisons may be limited by the lack of data on the quality variations in the clinical images used. Existing studies may not accurately reflect real-world outcomes, where clinical images captured by patients or in primary and secondary care settings often vary in quality and clarity, in contrast to high-resolution macroscopic or dermoscopic images taken by skilled dermatologists. [61]. 

Additionally, promising results have been observed with the use of automated algorithms for sequential digital dermoscopy (SDD) to support the early detection of malignant melanoma. AI can be trained to identify subtle differences between consecutive dermoscopic images, differences that might not be easily detected by the naked eye [57]. Hence, given the variability of macroscopic images in real-life applications as well as the additional benefits and specificity with dermoscopic images, the creation of an Asian dermoscopic image database will likely contribute to a better and more consistent training set for AI.

### 4.6. Future Research Directions

Moving forward, while we aim to maximize the transformative potential of AI, it is prudent to keep in mind the ethical and legal challenges that comes with it. Deep learning models require large amounts of data and image annotation for training and validation. This highlights the need for stringent quality control of images and AI models used to ensure that patient safety is not undermined. 

Accessing medical data poses privacy and legal concerns [62]. Unequal racial and ethnic representation in current databases used for training AI models can further exacerbate healthcare inequality. AI decisions should also be assessed for safety and transparency, as incorrect decisions can be fatal [63]. Undue reliance on AI, especially amongst the younger and less experienced clinicians, may risk compromising clinicians’ expertise with traditional diagnostic and treatment methods [64].

Future collaborations between institutions to construct a publicly available, validated image database made up of diverse Asian populations and skin lesion types will be crucial to mitigate biases, improve the robustness of deep learning algorithms, and allow us to better adapt results into real-life practice. Appropriate legal regulations should be implemented to facilitate the safe exchange of patient medical data between clinical centers and other scientific institutions [64]. More efforts should also be put into the development of prospective clinical trials to develop AI models that are capable of multimodal input to incorporate more information, just as a physician would, to arrive at a more accurate final diagnosis. Human–machine collaboration can be studied to identify loopholes and for constructive feedback on the training of the AI algorithms. Researchers and developers should also be cognizant of the target group that their AI model should benefit, be it patient-facing, primary care, or tertiary care settings, so that the design of AI algorithms can be tailored to suit the needs of the target audience. For instance, patient-facing AI technologies will need to be able to analyze and classify macroscopic images accurately with high specificity to avoid unnecessary referrals to specialist clinics.

### 4.7. Strengths and Limitations

The main strengths of our study lie in the extensive and systematic search in two different databases and strict inclusion criteria of only studies which utilized Asian databases for training or validation.

Limitations include the potential for selection bias, as articles from databases not included in this review may have been excluded. While the stricter selection criteria resulted in inclusion of fewer Asian studies for review, this approach helps minimize the risk of publication bias and misrepresentation as studies that rely on widely used public databases like ISIC and HAM10000, which predominantly feature Caucasian populations with lighter Fitzpatrick skin types, tend to be more statistically significant. These studies might also present risks of overlap in images used for training and testing, introducing further biases. Therefore, our approach enables a more comprehensive understanding of the AI models currently in use, their outcomes, and the limitations associated with existing Asian population databases.

The transferability of the results, particularly for skin cancers more prevalent in Asian populations, may also be uncertain since only around half of the studies used histology as the gold standard for diagnosis, coupled with a lack of quality control to ensure accurate and consistent annotation of images, sufficient database diversity, and external validation.

## 5. Conclusions

Overall, AI technologies have shown great potential to aid in the early detection and diagnosis of skin cancers in Asian populations. There has been a shift towards the adoption of deep learning models such as CNNs, but research generated using newer and more refined transformer models remains limited. The current literature lacks representation from Asian populations with darker Fitzpatrick skin type, as well as dedicated dermoscopic images or rare skin cancer databases.

Appropriate implementation of AI, guided by evidence-based approaches, is prudent to maximize its efficacy. Ongoing efforts should focus on diversifying and expanding Asian databases to encompass a wide range of Asian genotypes and skin conditions, while ensuring rigorous quality control standards are maintained. Future collaborations should focus on prospective, real-world studies to improve transferability of the current AI technologies developed in Asia and address potential ethical and legal challenges.

## Figures and Tables

**Figure 1 diagnostics-15-00939-f001:**
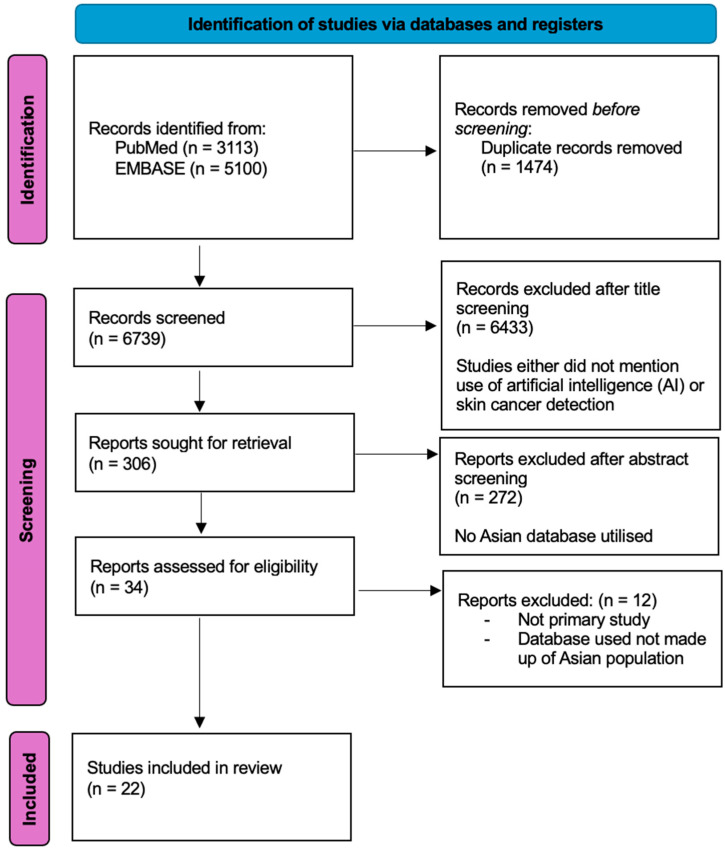
Summary of systematic review performed according to the preferred reporting items for systematic reviews and meta-analysis guidelines.

**Table 1 diagnostics-15-00939-t001:** Characteristics of studies (*n* = 22).

Characteristics	*n* (%)	Reference
Publication year		
2005	1	[31]
2008	1	[45]
2013	1	[32]
2017	2	[41,44]
2018	2	[24,29]
2019	2	[30,35]
2020	7	[26,27,28,34,36,39,43]
2021	2	[25,37]
2022	2	[33,40]
2023	1	[42]
2024	1	[38]
Country of publication		
China	8	[27,30,33,34,38,39,41,42]
South Korea	8	[24,25,26,28,29,36,40,44]
Japan	4	[35,37,43,45]
Taiwan	1	[32]
Iran	1	[31]
AI model employed		
Deep		
CNN	17	[24,25,26,27,28,29,30,33,34,35,36,37,38,39,40,42,43]
ANN	2	[31,41]
Shallow		
SVM	1	[32]
Linear Model	1	[45]
Machine learning (histogram)	1	[44]
Classification types		
Binary (benign vs. malignant)	16	[24,25,27,28,30,31,33,35,36,38,39,41,43,44,45]
Multiclass	12	[26,28,29,32,33,34,35,37,39,40,42,43]
Modality used		
Dermoscopy	9	[24,25,31,34,37,38,41,44,45]
Clinical	11	[26,27,28,29,30,32,33,35,36,40,43]
Clinical + Dermoscopy	2	[39,42]
Gold standard diagnosis of skin lesion included		
Histopathological	11	[24,25,26,27,31,33,34,35,38,39,44]
Mixture of Clinical consensus on diagnosis and histopathological confirmation	6	[28,36,37,40,43,45]
Not specified	5	[29,30,32,41,42]
Types of skin cancer evaluated		
Melanoma (all subtypes)	16	
Cutaneous melanoma	12	[26,29,31,32,33,35,36,37,39,40,41,43]
Acral melanoma	4	[24,25,44,45]
Basal cell carcinoma	15	[26,27,29,30,32,33,34,35,36,37,38,39,40,42,43]
Squamous cell carcinoma	7	[26,29,32,33,35,36,39,40]
Intraepithelial Ca	2	[26,29]
Kaposi’s sarcoma	1	[32]
Adenocarcinoma of lips	1	[36]
Data pre-processing		
Pre-processing performed	12	[25,27,30,32,33,35,36,37,38,39,41,43]
Pre-processing not performed/not specified	10	[24,26,28,29,31,34,40,42,44,45]
Data augmentation		
Performed	4	[24,25,33,35]
Not performed/specified	18	[26,27,28,29,30,31,32,34,36,37,38,39,40,41,42,43,44,45]

**Table 2 diagnostics-15-00939-t002:** Characteristics of AI models trained and datasets included.

Ref.	AI	AI Model	Tasks of AI Model	Dataset	Dataset Sample Size	Skin Cancers Included in Dataset	Classification Types	Dermoscopy/Clinical
[24]	CNN	VGG-16 model	Feature extraction and binary classification	Severance Hospital in Yonsei University Health System and Dongsan Hospital in Keimyung University Health System	350 acral melanoma, 374 benign naevus	AM	Binary (acral melanoma vs. benign naevus)	Dermoscopy
[25]	CNN	Deep ConvNet (custom-built) AlexNet ResNet-18	Feature extraction and binary classification	Yonsei University Health System South Korea	350 acral melanoma, 374 benign naevus	AM	Binary (acral melanoma vs. benign naevus)	Dermoscopy
[26]	CNN	Model Dermatology with RCNN (custom-built) Disease classifier (SENet and SE-ResNeXt-50) was trained with help of region-based CNN (faster RCNN)	Cropped-image analysis, blob detector and fine image selector of unprocessed images for detection, multiclass classification of diagnosis and calculation of malignancy outputs	Severance Hospital, Korea	10,426 (Severance dataset A for binary classification) 10,315 (Severance dataset B * for multiclass classification analysis)	MM, BCC, Intraepithelial Ca (SCC-in situ), SCC	Binary (benign vs. malignant) multiclass predict exact diagnosis of skin disease	Clinical
[27]	CNN	Inception V3, InceptionResNetV2, DenseNet121, ResNet50	Binary classification	Xiangya Hospital, Central South University, China	541 BCC, 684 SK	BCC	Binary (BCC vs. seborrheic keratosis)	Clinical
[28]	CNN	Custom CNN	Binary classification (predicting malignancy and suggesting treatment options), multiclass classification of skin disorders	SNU dataset, Korea	2201 (134 disorders)	NA	Binary multiclass	Clinical
[29]	CNN	Microsoft ResNet-152 (Trained with Asan, MED-NODE and atlas dataset)	Multiclass classification of cutaneous tumors	ASAN medical centre (> 99% asians) testing portion of dataset, Hallym Korea	1276 (Asan test set), 152 BCC (Hallym test set)	BCC, SCC, intra-epithelial Ca, MM	Multiclass	Clinical
[30]	CNN	Xception, ResNet50, InceptionV3, InceptionResNetV2, DenseNet121	Binary classification	Xiangya Skin Disease Dataset, China	349 BCC, 497 pigmented naevi	BCC	Binary (BCC vs. pigmented naevi)	Clinical
[31]	ANN	Visiomed AG (ver.350) based on ANN trained using images from Europe-wide multicenter study (DANAOS)	Binary classification (calculate likelihood of malignancy)	Iranian patients; Pakistan, Razi hospital	122 pigmented skin lesions	MM	Binary (benign vs. melanoma)	Dermoscopy
[32]	SVM	custom CAD	Feature extraction (shape, color, and texture), ranking of differentiating criteria, selection and multiclass classification	Kaohsiung Medical University, Taiwan	769 (174 malignant and 595 benign)—110 BCC, 8 MM, 14 Kaposi’s sarcoma, 20 SCC	MM, BCC, Kaposi’s sarcoma, SCC	Multiclass (a) exact diagnosis; (b) benign vs. malignant vs. indeterminate	Clinical
[33]	CNN	EfficientNet-B3	Binary and multiclass classification	Chinese PLA General Hospital & Medical School	25,773 used for training and validation; 2107 for testing (2178 BCC, 1108 SCC, 1030 MM)	BCC, SCC, MM	Binary multiclass (disease specific classification)	Clinical
[34]	CNN	GoogLeNet Inception v3	Multiclass classification	Peking Union Medical College Hospital, China	378 BCC	BCC	Multiclass	Dermoscopy
[35]	CNN	GoogLeNet DCCN	Binary and multiclass classification	Dermatology division of Tsukuba Hospital, Japan	6009 (some were different angles of same lesion), 4867 used for training, 1142 for testing	SCC, BCC, MM	1st level: Binary (benign vs. malignant) 3rd level: Multiclass	Clinical
[36]	CNN	Inception-Resnet-V2	Binary classification of lip disorders	Seoul National University Hospital (SNUH), Seoul National University Bundang Hospital, SMG-SNU Boramae Medical Center	1629 for training (743 malignant, 886 benign), 344 SNUH for internal validation, 281 SNUBH and SMG-SNNU for external validation	MM, BCC, SCC, AdenoCa over Lips	Binary (benign vs. malignant)	Clinical
[37]	CNN	Inception-ResNeet V2	Multiclass four disease classification	Shinsu database	594 training set (49 MM, 132 BCC), 50 test set (12 MM, 12 BCC)	MM, BCC	Multiclass (4 disease classifier)	Dermoscopy
[38]	Machine learning	Combination model developed from 207 machine learning models, integrating XGBoost combined with Lasso regression for analysis of data	Binary classification	DAYISET 1 (First Affiliated Hospital of Dalian Medical University) for external validation set	63 (32 BCC, 31 Actinic keratosis)	BCC	Binary (BCC vs. AK)	Dermoscopy
[39]	CNN	Youzhi AI software (Shanghai Maise Information Technology Co., Ltd., Shanghai, China) (GoogLeNet Inception v4 CNN used as basis)	Binary classification	China Skin Image Database (CSID), China–Japan Friendship Hospital, China	106 (4 MM, 5 SCC, 24 BCC)	MM, SCC, BCC	1st level: Binary (benign vs. malignant) 2nd level: Multiclass (14 types of skin tumors)	Clinical and dermoscopy
[40]	CNN	Model Dermatology (custom-built)—output values of SENet, SE-ResNeXt-101, SE-ResNeXt-50, ResNeSt-101, ResNeSr-50 arithmetically averaged to obtain a final model output	Multiclass classification	ASAN dataset, South Korea (and web dataset)	120,780 clinical images from ASAN dataset used for training; 17,125 subset of ASAN dataset used for validation	MM, SCC, BCC	Multiclass	Clinical
[41]	ANN	Neural network meta-ensemble model: combination of BP neural networks with fuzzy neural networks to increase individual net diversity	Segmentation, feature extraction and binary classification	Xanthous set from General Hospital of the Air Force of The Chinese People’s Liberation Army, China	240 images (80 malignant, 160 benign)	MM	Binary (benign vs. malignant)	Dermoscopy
[42]	CNN	Proposed MFF-Net (multi-scale fusion structure combines deep and shallow features within individual modalities to reduce the loss of spatial information in high-level feature maps) on EfficientNet as backbone in a single modality	Feature extraction, multiclass classifications	Peking Union Medical College Hospital, China	3853 image pairs	BCC	Multiclass (BCC vs. naevus, Seb K, wart)	Clinical and dermoscopy
[43]	CNN	FRCNN with VGG-16 as backbone	Binary and multiclass classification	National Cancer Center Hospital, Tokyo, Japan	5846 brown to black pigmented skin lesions (1611 MM, 401 BCC)	MM, BCC	Binary (benign vs. malignant) multiclass (6-class classification)	Clinical
[44]	Machine learning	custom algorithm (Gaussian derivative filtering + histogram fo width ratio to classify lesions according to ridge/furrow width ratio)	Binary classification	Severance Hospital at Yonsei University Health System, Korea	297 images (184 acral melanoma, 113 benign naevi)	AM	Binary (acral melanoma vs. benign naevus)	Dermoscopy
[45]	Machine learning	Linear classifier model (custom-built) (categorized into color, symmetry, border, and texture)	Automated tumor area extraction, Binary (melanoma–nevus and three pattern detectors) classification	Japanese hospitals (Keio University Hospital, Tornoman Hospital, Shinshu University Hospital, Inagi-Hospital) and two European universities as EDRA-CDROM database (only Japanese results included in this review); computer-based classification of dermoscopy images of melanocytic lesions on acral volar skin.	213 images (176 naevi, 37 melanomas)	AM	Binary (melanoma vs. benign naevus)	Dermoscopy

AM: acral melanoma; MM: malignant melanoma; SCC: squamous cell carcinoma; BCC: basal cell carcinoma. * angiosarcoma, dermatofibrosarcoma protuberance, Kaposi sarcoma, and Merkel cell carcinoma were excluded from dataset B used for multiclass analysis.

**Table 3 diagnostics-15-00939-t003:** Study outcomes.

Ref.	Evaluation Metrics (for AI Models and Clinicians)			Outcomes
	AUC	Sensitivity	Specificity	
[24]	CNN	CNN	CNN	Accuracy of CNN comparable to experts but surpasses non-experts
	(Group A) 83.51%	(Group A) 92.57%	(Group A) 75.39%	
	(Group B) 80.23%	(Group B) 92.57%	(Group B) 68.16%	
	(Average) 81.87%	(Average) 92.57%	(Average) 71.78%	
	Experts	Experts	Experts	
	(Group A) 81.08%	(Group A) 94.88%	(Group A) 68.72%	
	(Group B) 81.64%	(Group B) 98.29%	(Group B) 65.36%	
	Non-experts	Non-experts	Non-experts	
	(Group A) 67.84%	(Group A) 41.71%	(Group A) 91.28%	
	(Group B) 62.71%	(Group B) 48%	(Group B) 77.10%	
[25]	(Proposed ConvNet) 91.0%	NA	NA	CNN sensitivity is higher than human experts and can be used for early diagnosis of AM by non-experts. Transfer learning achieved better results than custom-built AI model.
	(ResNet-18) 97.5%			
	(AlexNet) 95.9%			
[26]	(CNN binary) 86.3%	(CNN Binary) 62.7%	(CNN Binary) 90%	Performances of algorithms were comparable to dermatologists in experimental setting but inferior to dermatologists in real-world practice. This could be due to the limited data relevancy and diversity involved in differential diagnoses in practice.
		(Clinician binary) 70.2%	(Clinician binary) 95.6%	
		(CNN multiclass): 66.9%	(CNN multiclass): 87.4%	
		(CNN BCC) 66.6%	(CNN BCC) 90%	
		(Clinician BCC) 74%	(Clinician BCC) 95.6%	
		(CNN SCC) 70.9%	(CNN SCC) 90%	
		(Clinician SCC) 65.8%	(Clinician SCC) 95.6%	
		(CNN MM) 61.4%	(CNN MM) 90%	
		(Clinician MM) 68.7%	(Clinician MM) 95.6%	
[27]	Trained from scratch	Trained from scratch	Trained from scratch	InceptionResNetV2 outperformed average of 13 general dermatologists, comparable to average of 8 expert dermatologists in ability of clinical image classification. Transfer learning achieved better results than custom-built AI model.
	(Inception V3) 89.4%	(Inception V3) 85.2%	(Inception V3) 84.6%	
	(Inception ResNetV2) 89.5%	(Inception ResNetV2) 85.4%	(Inception ResNetV2) 85.9%	
	(DenseNet 121) 89%	(DenseNet 121) 84.6%	(DenseNet 121) 84.8%	
	(ResNet50) 87.9%	(ResNet50) 78.3%	(ResNet50) 88.3%	
	Fine-tuned with ImageNet	Fine-tuned with ImageNet	Fine-tuned with ImageNet	
	(Inception V3) 89.6%	(Inception V3) 85.2%	(Inception V3) 83.7%	
	(Inception ResNetV2) 91.9%	(Inception ResNetV2) 79.1%	(Inception ResNetV2) 91.5%	
	(DenseNet 121) 91.3%	(DenseNet 121) 88.5%	(DenseNet 121) 81.6%	
	(ResNet50) 90.5%	(ResNet50) 80.8%	(ResNet50) 89.4%	
[28]	(CNN Binary) 93.70%	NA	NA	CNN showed similar performance as dermatology residents but slightly lower than dermatologists.
	(CNN multiclass) 97.8%			
[29]	Asan dataset	Asan dataset	Asan dataset	Varied subtypes in BCC among different ethnic groups can explain the lower AUC obtained by AI model trained with Asian dataset when validated in an Edinburgh test set.
	(BCC) 96%	(BCC) 88.8%	(BCC) 91.7%	
	(SCC) 83%	(SCC) 82%	(SCC) 74.3%	
	(MM) 96%	(MM) 91%	(MM) 90.4%	
	(Intraepithelial Ca) 82%	(Intraepithelial Ca) 77.7%	(Intraepithelial Ca) 74.9%	
		Hallym dataset		AI showed comparable performance to 16 dermatologists in ability to classify 12 skin tumor types and was superior in diagnosis of BCC.
		(BCC) 87.1%		
[30]	(InceptionV3):	(InceptionV3):	(InceptionV3):	Xception had the best performance out of the five mainstream CNNs. Ability of Xception model to identify clinical images of BCC and nevi was comparable to that of professional dermatologists.
	94.4%	93.4%	88.1%	
	(ResNet50):	(ResNet50):	(ResNet50):	
	91.9%	86.3%	89.7%	
	(InceptionResNetV2):	(InceptionResNetV2):	(InceptionResNetV2):	
	96.3%	90%	93.6%	
	(DenseNet121):	(DenseNet121):	(DenseNet121):	
	96.3%	90.8%	93.2%	
	(Xception):	(Xception):	(Xception):	
	97.4%	94.8%	93.0%	
[31]	NA	(Melanoma) 83%	(Melanoma): 96%	Diagnostic accuracy of the computer-aided dermoscopy system was at the level of clinical examination by dermatologists with naked eyes and can help to reduce unnecessary excisions or improve early melanoma detection. It serves to improve diagnostic accuracy of inexperienced clinicians in evaluation of pigmented skin lesions.
[32]	(CADx) 90.64%	(CADx) 85.63% (Physician) 83.33%	(CADx) 87.65% (Physician) 85.88%	CADx performed similarly to that of dermatologists in ability to classify both melanocytic and non-melanocytic skin lesions by utilizing conventional digital macrophotographs.
	BCC: 90%			
	MM: 75%			
	Kaposi’s sarcoma: 71.4%			
	SCC: 80%			
	(Physician) 85.31%			
	BCC: 88.1%			
	MM: 75%			
	Kaposi’s sarcoma: 78.5%			
	SCC: 85%			
[33]	CNN	Binary (dermatologists without CNN assistance vs. with assistance): Sn 83.21% vs. 89.56%	Binary (Dermatologists without CNN assistance vs. with assistance) Sp 80.92% vs. 87.90%	Performance of dermatologists improved with CNN assistance especially for lesions with similar visual appearances.
	General top diagnosis: 78.45%			
	BCC: 78.0%			
	SCC: 91%			
	MM: 87%			
	Dermatologists without CNN assistance			
	General: 62.78%			
	BCC 55%			
	SCC 64%			
	MM 65%			
	Dermatologists with CNN assistance			
	General: 76.6%			
	BCC 71%			
	SCC 82%			
	MM 82%			
	CNN			
[34]	CNN Multiclass average: 81.49% BCC: 97.2%	CNN BCC: 80% Dermatologists BCC: 77%	CNN BCC: 100% Dermatologists BCC: 96.2%	Based on a single dermoscopic image, the performance of CNN was comparable to 164 board-certified dermatologists in the classification of skin tumors.
[35]	CNN	CNN Binary/1st level classification: 96.30%	CNN Binary/1st level classification: 89.50%	Trained DCNN could classify skin tumors more accurately than board-certified dermatologists (BCD) on basis of a single clinical image.
	Binary/1st level classification: 93.4%			
	Multiclass/3rd level classification:			
	General: 74.5%			
	SCC: 82.5%			
	BCC: 80.3%			
	MM: 72.6%			
	Dermatologists			
	Binary/1st level classification:			
	(BCD): 85.3%			
	(Dermatolgy trainees): 74.4%			
	Multiclass/3rd level classification:			
	BCD vs. trainees: 59.7% vs. 41.7%			
	(BCD)			
	SCC: 59.5%			
	BCC: 64.8%			
	MM: 72.8%			
[36]	(DCNN test set): 82.7%	(DCNN Test set): 75.5%	(DCNN test set): 80.3%	Specificity of the algorithm at the dermatologist’ mean sensitivity was significantly higher than human readers (*p* < 0.001). Sensitivity and specificity of dermatology residents, non-specialist and medical students improved after referencing DCNN output. But sensitivity and specificity of board-certified dermatologist did not have any significant improvement.
	(DCNN non-SNUH test set): 77.4%	(DCNN non-SNUH test set):70.2%	(DCNN non-SNUH test set): 75.9%	
	(DCNN combined): 81.1%	(DCNN combined): 73.7%	(DCNN combined): 77.9%	
[37]	Deep Neural Network	Dermatologist (Shinshu set): 85.30%	Deep Neural Network (Shinshu set): 96.2% Dermatologist (Shinshu set): 92.20%	DNN diagnostic performance with vs. without training >sn: improved from 0.875 > 0.917 >accuracy of malignancy prediction: improved from 0.920 > 0.940 >sp: did not change at 0.962 Dermoscopic diagnostic performance of Japanese dermatologists for skin tumors diminished for patients of non-local populations, particularly in relation to the dominant skin type. DNN may help close this gap in clinical settings.
	(Shinshu set, MM): 75%			
	(Shinshu set, BCC): 75%			
	General AUC: 94%			
	Dermatologist			
	(Shinshu set, MM) 65%			
	(Shinshu set, BCC) 80%			
[38]	(DAYISET 1/DATASET 4): 63.4%	(DAYISET BCC): 68.8%	NA	The model demonstrated high accuracy in discrimination and diagnosis of BCC and AK and can assist less experienced dermatologists in distinguishing skin lesions.
[39]	Binary:	Binary: AI (clinical + dermoscopic): 74.8% AI (clinical): 71.1% AI (dermoscopic): 78.6%	Binary: AI (clinical + dermoscopic): 93.0% AI (clinical): 90.6% AI (dermoscopic): 95.3%	No statistical difference in the diagnostic accuracy of Youzhi AI software and dermatologist under the two modes. Diagnostic accuracy of the Youzhi AI software was reduced in practical work possibly because dermoscopic images collected in clinical work tend to be less typical, increasing diagnostic difficulty for dermatologist and software.
	AI (Dermoscopic + clinical) 85.6%			
	AI (dermoscopic) 88.7%			
	AI (clinical) 83.0%			
	Dermatologists (dermoscopic + clinical) 83.3%			
	Dermatologists (dermoscopic): 89.6%			
	Dermatologists (clinical) 79.5%			
	Multiclass classification			
	AI (dermoscopic + clinical): 73.1%			
	AI (dermoscopic): 76.4%			
	AI (clinical): 68.9%			
	Dermatologists (dermoscopic + clinical): 61.4%			
	(dermoscopic): 63.4%			
	(clinical): 59.4%			
[40]	AI: BCC: 53.3% MM: 66.7% SCC: 50%	Sensitivity (based on top-3 predictions for malignancy determination) - AI assisted group; 84.6% - unaided group: 75.6%	NA	Multiclass AI algorithm augmented the diagnostic accuracy of non-expert physicians in dermatology.
	Top-1 accuracy (AI standalone): 50.5% (AI augmented group vs. unaided in general): 53.9% vs. 43.8% (AI_GP_ group vs. unaided_GP_): 54.7% vs. 29.7% (AI_resident_ group vs. unaided_resident_): 53.1% vs. 57.3%			
	Top-1 accuracy performance BCC: - (AI) 53.3% - (Trainee) 33.3% > 60% (with augmentation) - (Attending) 60%			
	Melanoma - (AI) 66.7% - (Trainee) 83.3% > 50% (with augmentation) - (Attending) 83.3%			
	SCC - (AI) 50% - (Trainee) 62.5% > 50% (with augmentation) - (Attending) 62.5%			
[41]	94.17%	95%	NA	The proposed lesion border features are particularly beneficial for differentiating malignant from benign skin lesions
[42]	EfficientNet-B4 without multi-scale integration (BCC) (Dermoscopic): 78.4% (Clinical): 84.5% (Dermoscopic + clinical): 74%	NA	NA	The multi-scale fusion structure integrating the features of various scales within a single modality demonstrates the significance of intra-modality relations between clinical images and dermoscopic images.
	EfficientNet-B4 with multi-scale (BCC) (Dermoscopic): 80.1% (Clinical): 83% (Dermoscopic + clinical): 76.2%			
[43]	FRCNN (MM): 80.1% (BCC): 81.8% (Binary): 91.5%	FRCNN: 83.3% BCDs: 86.3% TRNs: 85.3%	FRCNN: 94.5% BCDs: 86.6% TRNs: 85.9%	Similar in performance between FRCNN and BCDs but slightly better than TRNs for specific diagnosis. But for differentiation between malignant vs. benign—FRCNN had highest accuracy.
	Board-certified dermatologists (BCDs) (MM) 83.3% (BCC) 78.8% (Binary) 86.6%			
	Trainees (TRN) (MM) 80.1% (BCC) 65.9% (Binary) 85.3%			
	For multiclass (six classes) classification, FRCNN had higher accuracy rates than BCD and TRN (FRCNN) 86.2% (BCDs) 79.5% (TRNs) 75.1%			
[44]	99.70%	100%	99.1%	The proposed algorithm agrees with dermatologists’ observations and achieved high accuracy rates.
[45]	93.30%	93.3%	91.1%	NA
	(MM, images) 80.6%			

‘Sn’ denotes sensitivity; ‘Sp’ denotes specificity; ‘NA’ denotes not applicable.

## Data Availability

The data used in this systematic review were primarily obtained from publicly accessible databases of PubMed and EMBASE. However, some studies required additional data access through contacting the original authors. Due to data-sharing restrictions imposed by certain studies, not all data analyzed are readily available. Requests for access to specific datasets should be directed at the relevant study authors where possible.

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
