# Peer review of "The Use of Artificial Intelligence for Skin Cancer Detection in Asia—A Systematic Review"

_diagnostics, 2025, doi:10.3390/diagnostics15070939_

Round 1
Reviewer 1 Report
Comments and Suggestions for Authors
The article is a systematic review of the use of AI to detect skin cancer in Asia. The study highlights the applicability of existing AI models to Asian populations and the inadequacy of existing datasets.
The study analyzed 22 studies by conducting a comprehensive search of PubMed and EMBASE databases. The geographical distribution of the studies, the AI ​​models used, and the data sources were examined in detail. A comparative evaluation of transfer learning and AI models was performed. Different methods (CNN, ANN, SVM, etc.) were compared.
The study provided very useful analyses when evaluated as a review study. The following corrections would also improve the quality of the study.
- Most of the studies focus on China, Japan, and South Korea. Indian, Southeast Asian, and Middle Eastern populations with darker skin tones are underrepresented. It would be useful to add a few studies from those regions.
- A more detailed comparison should be made between different AI models. For example, Transformer-based models (ViT, Swin Transformer) vs. CNN models.
- The article did not touch on practical aspects such as the running time of AI models, hardware requirements, and inference speed. It would be good if it added this data as well.
Reviewer 2 Report
Comments and Suggestions for Authors
Review Report for MDPI Diagnostics
(The Use of Artificial Intelligence for Skin Cancer Detection in Asia – A Systematic Review)
1. Within the scope of the study, studies on the use of artificial intelligence in skin cancer detection studies in Asia have been examined in depth.
2. In the introduction section, skin cancer and the use of artificial intelligence are mentioned at a basic level. At the end of this section, the differences and main contributions of this study from other similar review papers in the literature should be stated clearly and in bullet points.
3. In the Materials and Methods section, it is mentioned that the publications related to the study were obtained from EMBASE and PubMed databases. The keywords mentioned here seem sufficient.
4. Adding the titles of data preprocessing and/or data augmentation as an additional section in Table-1 and considering the articles from this perspective will increase the quality of the study.
5. In Table-2, AI models, dataset and classification types are clearly stated, which proves that in-depth analysis was performed within the scope of the study.
6. In Table-3, the analyses performed in the studies in terms of evaluation metrics are appropriate and guide similar studies in the future.
7. The positive aspects and shortcomings of artificial intelligence in relation to skin cancer detection studies in Asia are discussed clearly.
In conclusion, the study has the potential to make a significant contribution to the literature regarding skin cancer detection with artificial intelligence. However, attention should be paid to the sections listed above.
Reviewer 3 Report
Comments and Suggestions for Authors
Skin cancers are a significant threat that has been growing in recent times. Supporting dermatologists with AI methods in the diagnosis of skin lesions has become common due to the increasing clinical effectiveness of these methods. Therefore, the subject of this work is important and timely. However, some issues require clarification and explanation before it is suitable for publication.
1. Please explain in more detail how skin cancers in the Asian population differ from those of other populations. Asia is a large continent, encompassing many inhabitants. Can we assume that skin lesions of these inhabitants form a homogeneous, distinct group that should be analyzed separately? Especially since two countries, China and Korea, are overrepresented in Table 2.
2. Table 3 should be supplemented with information on what task the AI algorithms performed: classification (multi-class), segmentation, detection of moles, etc.
3. This work should be considered in the literature review: 10.18231/j.ijced.2025.001
4. The discussion should be expanded to include the following aspects:
- assessment of the effectiveness of AI algorithms in detecting/diagnosing neoplastic lesions compared to expert physicians
- the results of analyses obtained for Asian countries should be compared with those for other population groups (Europe, North America). For this purpose, published review papers should be used:
10.3390/life14121602
10.1049/ipr2.13219
10.1016/j.clindermatol.2023.12.022
10.3390/cancers15041183
- ethical and legal aspects of using AI in dermatology
Round 2
Reviewer 3 Report
Comments and Suggestions for Authors
Thank you for correctly addressing most of my comments.
However, the discussion did not include all the suggested publications. This should be done and the discussion should be supplemented, then the work will be suitable for publication.